# Characterization of Schistosome Sox Genes and Identification of a Flatworm Class of Sox Regulators

**DOI:** 10.3390/pathogens12050690

**Published:** 2023-05-09

**Authors:** Stephanie Wood, Kenji Ishida, James R. Hagerty, Anida Karahodza, Janay N. Dennis, Emmitt R. Jolly

**Affiliations:** 1Department of Biology, Case Western Reserve University, Cleveland, OH 44106, USA; smw116@case.edu (S.W.); kishida@childrensnational.org (K.I.); jamesrna@uchicago.edu (J.R.H.);; 2Center for Global Health and Disease, Case Western Reserve University, Cleveland, OH 44106, USA

**Keywords:** SoxB, Sox, *Schistosoma*, flatworms, helminths

## Abstract

Schistosome helminths infect over 200 million people across 78 countries and are responsible for nearly 300,000 deaths annually. However, our understanding of basic genetic pathways crucial for schistosome development is limited. The sex determining region Y-box 2 (Sox2) protein is a Sox B type transcriptional activator that is expressed prior to blastulation in mammals and is necessary for embryogenesis. Sox expression is associated with pluripotency and stem cells, neuronal differentiation, gut development, and cancer. Schistosomes express a Sox-like gene expressed in the schistosomula after infecting a mammalian host when schistosomes have about 900 cells. Here, we characterized and named this Sox-like gene *SmSOXS1*. SmSoxS1 protein is a developmentally regulated activator that localizes to the anterior and posterior ends of the schistosomula and binds to Sox-specific DNA elements. In addition to SmSoxS1, we have also identified an additional six Sox genes in schistosomes, two Sox B, one SoxC, and three Sox genes that may establish a flatworm-specific class of Sox genes with planarians. These data identify novel Sox genes in schistosomes to expand the potential functional roles for Sox2 and may provide interesting insights into early multicellular development of flatworms.

## 1. Introduction

Schistosome helminths infect over 200 million people across 78 countries in Asia, South America, and Africa and are responsible for nearly 200,000 deaths annually. Infection occurs when transient free-swimming cercarial larvae penetrate human skin and initiate a complex process to adapt to survival in the definitive vertebrate host. As the cercaria invades host skin, the cercarial tail is lost, initiating a complex series of developmental changes as the free-swimming cercaria transforms into a schistosomulum. This includes exchanging the glycocalyx surface with a syncytial tegumental surface to adapt from fresh water to saline blood [1]. In addition, the schistosomulum expands its primordial gut, elongates the body, undergoes muscle development, and adapts to evade the host immune response. Although our knowledge of schistosome parasites has expanded in recent years, our understanding of the genetic mechanism regulating early schistosome development has been relatively belated. In many organisms, embryonic development requires the expression of a host of embryonic genes, such as Forkhead box (Fox) genes [2], fibroblast growth factor (FGF) [3], and the sex determining region Y-box genes (Sox) family of transcription factors [4,5]. While these genes are primarily expressed in embryogenesis, several of these gene families are expressed in the extra-embryonic schistosomulum, made up of more than 900 cells [6].

The sex determining region Y-box (Sox) gene family is important for stem cell maintenance and early development in mammals, fish, and insects [5]. Sox proteins are classified into classes A–J historically based on the homology of the high mobility group (HMG) domain [5]. The HMG domain is conserved in the Sox gene family and is necessary for binding DNA in order to regulate expression of transcriptional targets. The Sox2 protein, a Sox B type transcriptional activator and one of the Yamanaka factors, is associated with pluripotency and stem cells, neuronal differentiation, gut development, and cancer [7]. In mammals, it is expressed prior to blastulation with other Sox genes and is necessary for embryogenesis [8,9].

While Sox proteins are primarily expressed during early development, schistosomes are unusual in that they express a Sox2-like gene after infecting their mammalian host, long after the formation of a blastula. In mammals, Sox2 proteins are typically found in precursor cells to maintain cell stemness and control important embryonic processes, including neurogenesis, pluripotency, inner ear development, mitosis, taste bud development, and other developmental processes [5,10,11,12,13,14,15,16,17,18,19]. Sox2 function during embryogenesis has primarily been explored in vertebrates, and there has been some characterization of Sox2 function in the invertebrate roundworm *Caenorhabditis elegans* and the insect *Drosophila melanogaster* [11,20,21]. In *C. elegans* and *D. melanogaster*, Sox homologs are important in neurogenesis and development of the central nervous system [13,20]. In *D. melanogaster*, the Sox homolog, SoxNeuro, plays a role in early neurogenesis and the formation of neuroectodermal progenitors and neuroblasts [11]. SoxBs and SoxCs are also important for early neurogenesis in *C. elegans*. In *C. elegans*, sox-2 is needed to maintain the blast cell potential for later differentiation into neuronal cells [13]. However, despite the importance of flatworms in neglected tropical diseases, little effort has focused on the identification or characterization of flatworms’ Sox proteins.

Here, we identified and characterized a Sox homolog in the flatworm *Schistosoma mansoni* that we have named *SmSOXS1*. We showed that *SmSOXS1* is expressed in embryonic sporocysts, cercariae, and 4 hour (4 h) schistosomula but not in adult worms. Sox proteins are often known to function as transcription factors. As Sox proteins are often transcriptional activators, we further demonstrated that SmSoxS1 can function as a transcriptional activator and is able to bind to Sox-specific DNA binding sequences. Using immunohistochemistry, we also demonstrated that the Sox protein (SmSoxS1p) is localized to the anterior tip of the developing schistosomula. Finally, we identified an additional six Sox family proteins in the schistosome genome, classifying a novel group of flatworm-specific Sox gene regulators distinct from the mammalian Sox lineage.

## 2. Materials and Methods

### 2.1. Animals and Parasites

Biomphalaria glabrata snails (NMRI strain) infected with *Schistosoma mansoni* (NMRI strain, NR-21962) were obtained from the Biomedical Research Institute (BRI, Rockville, MD, USA, MD_NIAID Schistosomiasis Resource Center under NIH-NIAID Contract HHSN272201700014I). *S. mansoni* cercariae were collected and mechanically transformed into schistosomula as previously described [22]. Cercarial heads and tails were separated by slow sedimentation [23]. The resulting schistosomula were maintained in culture in complete Dulbecco’s Modified Eagle Medium (DMEM) (Gibco, Waltham, MA, USA).

### 2.2. Bioinformatic Analysis

A Basic Local Alignment Search Tool (BLAST) (NCBI) using Sox2 protein sequence from mice (NCBI, accession number NP_035573) was used to screen for Sox2 homologs in the *S. mansoni* protein database using the SchistoDB (schistodb.net, accessed on 7 July 2011) and WormBase ParaSite (parasite.wormbase.org, accessed on 20 February 2021) websites to search for Sox homologs [24,25,26,27,28]. Schistosome Sox homologs were compared against Sox homologs from *Mus musculus* (Mm), *Drosophila melanogaster* (Dm), *Caenorhabditis elegans* (Ce), *Danio rerio* (Dr), and *Schmidtea mediterranea* (Smd) (www.ncbi.nlm.nih.gov, accessed on 20 February 2021) [29]. All NCBI Accession numbers and Smp numbers are listed in Appendix A. The HMG domain was determined using Expasy ScanProsite (Swiss Institute of Bioinformatics, https://prosite.expasy.org/scanprosite/, accessed on 23 April 2023) [30]. The HMG domain sequences were aligned using MUSCLE (v3.8.31, https://drive5.com/muscle/downloads_v3.htm, accessed on 23 April 2023) and GBlocks (v0.91b, Castresana, J., https://www.biologiaevolutiva.org/jcastresana/Gblocks.html, accessed on 23 April 2023) [31,32,33]. The phylogenetic tree was generated using MrBayes (v3.2.7, NBISweden, https://github.com/NBISweden/MrBayes/, accessed on 23 April 2023), with 3,000,000 generations, and visualized using FigTree (v1.4.4, Raumbaut, A., https://github.com/rambaut/figtree/, accessed on 23 April 2023) [34,35,36,37].

### 2.3. Molecular Cloning

The schistosome Sox gene Smp_301790 (*SmSOXS1*) was cloned from cDNA produced from 4 h schistosomula using forward primer oAK003 (5′-GGC CGA ATT CCC GGG GAT CCT GAT GCA ATC CAA TTT AGC TAA TAA TCA TT-3′) and reverse primer oAK004 (5′-CGC TGC AGG TCG ACG GAT CCT TAG AAA AAT GTT TGT AAA TCA ATG GAA TCT AA-3′) to produce a 900 base-pair PCR product. The PCR product was subcloned into the BamHI site of vector pGBKT7 (Clontech, Mountain View, CA, USA) to make plasmid pEJ1569, which was subsequently sequenced for verification.

### 2.4. Modified Yeast 1-Hybrid System

The yeast strain AH109 (Clontech, Mountain View, CA, USA) was transformed with plasmid pEJ1569 and plated on synthetic dextrose medium without tryptophan (SD-Trp). Transformed colonies were patched onto SD-Trp plates overlaid with 660 µg 5-bromo-4-chloro-3-indolyl-α-D-galactopyranoside (X-α-Gal), incubated at 30 °C and screened for blue color. In a secondary screen, yeast cells were serially diluted from 1 to 10^−5^ and selected for growth on synthetic dextrose medium without histidine and containing 2.5 mM 3-AT (SD-His 2.5 mM 3-AT). In the dilution series, “1” is equal to 2,000,000 cells.

### 2.5. RNA Purification

Cercaria were collected, suspended in TRIzol reagent, and homogenized by bead beating. Schistosomula were collected at 4 hours (4 h), 24 hours (24 h), 48 hours (48 h), and 72 hours (72 h) time points post transformation into schistosomula. Phenol-chloroform extraction was performed on the samples using a Clean and Concentrator Kit (Zymo, Irvine, CA, USA) following manufacturer’s guidelines, including an on-column DNase I treatment. The process was repeated for adult worms and uninfected *B. glabrata* snail RNA received from the BRI. Sporocyst RNA was extracted from infected snails using the PureLink RNA Mini Kit (Invitrogen, Waltham, MA, USA) using manufacturer’s guidelines, with an on-column DNase I treatment.

### 2.6. Reverse Transcription and Quantitative PCR

RNA reverse transcription was performed using SuperScript First-Strand Synthesis System for RT-PCR (Invitrogen, Waltham, MA, USA) with 500 ng RNA for each stage, with a no-RT control. Absolute qPCR (Applied Biosystems, Waltham, MA, USA), using 25 ng of the cDNA for each reaction, was performed in triplicate using primers oSW017 (5′-AGT TTT ATG TCC TAC CCG TTC AAA-3′) and oSW018 (5′-GGT TCT GGC TTA TGG TTC ATC TC-3′). A linearized plasmid, containing *SmSOXS1*, was used as a standard. Uninfected *B. glabrata* snail RNA was used as a negative control for the sporocyst RNA.

### 2.7. Recombinant Protein Expression and Protein Purification and Parasite Lysate Preparation

Recombinant SmSoxS1 was produced, expressed, and purified as a Fusion protein with Maltose binding protein (MBP) using the pMAL Protein Fusion and Purification System (NEB, Ipswitch, MA, USA). *SmSOXS1* was subcloned into pMAL-c5X (NEB, Ipswitch, MA, USA), and this plasmid construct was transformed and induced in BL21 (DE3) *E. coli* cells (Invitrogen, Waltham, MA, USA) with 0.4 mM IPTG. Cells were disrupted by sonication in lysis buffer (50 mM potassium phosphate pH 8.0, 200 mM sodium chloride; Halt protease inhibitor (Thermo Scientific, Waltham, MA, USA); PMSF). Cleared supernatant was incubated with amylose–agarose beads (NEB, Ipswitch, MA, USA) with agitation overnight at 4 °C. The fusion protein was eluted from the beads using a maltose elution solution (50 mM potassium phosphate pH 8.0, 200 mM sodium chloride, 10.4 mM maltose) and quantified using the BCA Protein Quantitation kit (Thermo Scientific, Waltham, MA, USA). Schistosomula samples were disrupted by bead beating with glass beads (Sigma, Burlington, VT, USA) and sonication in a Tris-HCl lysis Buffer (25 nM Tris-HCl pH 7.5, 1 mM DTT, 1X Halt Protease Inhibitor, 1 mM PMSF). Protein was quantified using Pierce BCA Protein Assay Kit (Thermo Scientific, Waltham, MA, USA).

### 2.8. Electrophoretic Mobility Shift Assay (EMSA)

A LightShift Chemiluminescent EMSA Kit (Thermo Scientific, Waltham, MA, USA) was used to test SmSoxS1 DNA binding using manufacturer’s instructions. Probes were designed based on the consensus binding motif for Sox family proteins (5′-CWTTGWW-3′). These probes included sequences from *SmSOXS1*, with the consensus binding motif underlined, (GenBank accession number HE601624, 5′-TAT CAA TTT GAC GTT ACA TTG ATG GTT TAT TGT GAT GGG-3′), *SmFGF4* (Smp_035730, GenBank accession number HE601627; 5′-AGC TTC TCG AGC TGC TCT TTG TTT GGT TAC TCT AAA TAC-3′), and NDT80 (described previously as AT014; 5′-GCC GAT CCG CAT TTT GTG ACA TCC TTC AGC-3′) [26,38,39]. Biotinylated “hot” DNA probes and non-biotinylated “cold” DNA probes were obtained (IDT, Coralville, IA, USA) and annealed (99 to 30 °C, −1 °C/min). The probes and protein were mixed with 0.75 µg poly dI·dC and incubated for 30 minutes, followed by separation on a 5% native polyacrylamide (0.5 × TBE) gel and transferred onto Biodyne B Pre-Cut Modified Nylon Membrane (Thermo Scientific, Waltham, MA, USA). The membrane was probed using streptavidin-HRP and signal was detected using Amersham ECL Prime Western Blotting Detection Reagent (GE Healthcare, Chicago, IL, USA).

### 2.9. Western Blot Analysis

Further, 10 µg of protein from 4 h and 24 h schistosomula and 50 ng of MBP and MBP–SmSoxS1 protein were used for Western analysis. The custom SmSoxS1-1 primary antibody (GenScript, Piscataway, NJ, USA) was diluted to 0.68 µg/mL in 5% milk-PSBTw (PBS/0.1% Tween-20) solution and divided into primary antibody and peptide block solution. The peptide block solution contained the peptide for the SmSoxS1-1 antibody at a 10× concentration by weight. An HRP-link goat anti-rabbit secondary antibody (GE Healthcare, Chicago, IL, USA) was used at a 1:2000 dilution in 1% milk-PBSTw and visualized using Amersham ECL Primer Western Blotting Detection Reagent (GE Healthcare, Chicago, IL, USA). The membranes were exposed to autoradiography film and developed for 10 min.

### 2.10. Immunohistochemistry

A protocol adapted from Collins et al. was used to prepare samples for immunohistochemistry [40]. Briefly, 4 h and 24 h schistosomula were fixed for 20 min at room temperature in a 4% paraformaldehyde/PBSTw (PBS/0.1% Tween-20) solution, washed in PBSTw, then dehydrated in a methanol/PBSTw series and stored in 100% methanol at −20 °C until use. Prior to use, schistosomula were rehydrated, digested for 10 min at room temperature in permeabilization solution (1 × PBSTw, 0.1% SDS, and proteinase K (1 µg/mL)), and washed in PBSTw (all subsequent washes were carried out with nutation at room temperature). Schistosomula were re-fixed for 10 min at room temperature in a 4% paraformaldehyde/PBSTw solution and washed in PBSTw. Samples were incubated with rocking in block solution (PBSTw, 5% horse serum (Jackson ImmunoResearch Laboratories, West Grove, PA, USA), 0.05% Tween-20, and 0.3% Triton X-100) for 2h at RT. Samples were incubated with a polyclonal primary rabbit anti-SmSoxS1-1 antibody in block solution at a concentration of 2.5 µg/mL overnight at 4 °C and went through seven 20-min washes at room temperature. Samples were then incubated with an Alexa 647 donkey anti-rabbit antibody (Jackson ImmunoResearch Laboratories, West Grove, PA, USA) at a concentration of 1:600 in block solution overnight at 4 °C and washed in PBSTw 4 times for 20 min at room temperature with the second wash containing DAPI (1 µg/mL). After washing, samples were mounted in Slow Fade Gold (Invitrogen, Waltham, MA, USA). The peptide block control, containing the peptide for the SmSoxS1-1 antibody at a 10× concentration by weight, and no primary controls were run in parallel with experimental samples.

### 2.11. Imaging

All images were taken using Leica SP8 confocal system (Leica Microsystems, Deerfield, IL, USA) with tunable white light laser. Leica HC PL APO CS2 63×/1.40 OIL and HCX PL APO 100×/1.40 OIL objectives used with a maximum optical zoom factor of 3.5×. Alexa 647 conjugated secondary was excited using 647 nm light and detector set at peak absorbance. DAPI was excited at 405 nm using UV diode. All images were analyzed and processed using Image J (v1.50, https://imagej.net/ij/index.html, accessed on 23 April 2023) and Leica LASX software (Leica Microsystems, Deerfield, IL, USA) [41].

## 3. Results

### 3.1. Schistosomes Encode a SoxB-like Gene

Sox proteins are members of the high mobility group (HMG) family of DNA binding proteins and are found throughout the animal kingdom [21,42]. While Sox sequences can be quite variable, the HMG domain sequence in the Sox family is highly conserved [11,21,42]. Sox2 proteins are transcriptional activators that regulate gene expression of several targets during embryonic development and are 98% identical in humans and mice across the entire protein [5,43,44]. We initially investigated a homolog of the Sox2 transcription factor in parasitic schistosomes (Smp_301790) based on its significant homology to human Sox2 (NP_003097) and the mouse Sox2 (NP_035573) using two databases: NCBI and SchistoDB [24,25,26,29]. The 300 amino acid Sox2 sequence from *Schistosoma mansoni* was 71% identical in sequence to the mouse Sox2 protein in the first 100 amino acids. These first 100 amino acids contain the conserved HMG domain in both mouse Sox2 and in schistosome Sox2 (Appendix A) [5,21,45]. Sox proteins are classified based on the sequence of the HMG domain, with Sox proteins from the same group being at least 70% identical [5,11,21]. Since the HMG domain of Smp_301790 is 71% identical to the mouse Sox2 protein, we initially classified Smp_301790 as a potential Sox2 homolog, which we tentatively named SmSoxS1.

### 3.2. SmSOXS1 Is Differentially Expressed

We cloned *SmSOXS1* from a mixture of cDNA extracted from sporocysts, mixed adults, and cercariae. The sequence analysis of the cloned gene mirrored the DNA sequence in the schistosome database: SchistoDB [24,25,26]. We then assessed the expression profile of *SmSOXS1* using absolute quantitative PCR in the infected snail host (5-week daughter sporocysts), the free-swimming cercarial stage, and in mammalian stages (4 h to 72 h schistosomula and adult worms). *SmSOXS1* had increasing transcript levels during the transition from sporocyst to cercaria and from cercariae to 4 h schistosomula, with a subsequent decrease over the course of three days as schistosomula (Figure 1). *SmSOXS1* transcript was not present in adult worms (Figure 1). When we compared the transcript levels to a known standard, cyclophilin (Smp_054330), we found that *SmSOXS1* has relatively low copy numbers, with around 4700 copies at its highest expression levels in 4 h schistosomula, while cyclophilin has around 38,000 copies at the same time point (Figure 1 and Appendix A) [38].

### 3.3. SmSoxS1 Is a Transcriptional Activator

Sox proteins generally function as transcriptional regulators, and a majority of Sox proteins, including mouse Sox2, function as activators [11,14,45]. To test whether SmSoxS1 protein could also function as an activator, we made a fusion protein with the DNA binding domain of the yeast Gal4 protein (Gal4-DBD) and SmSoxS1 (Gal4DBD-SmSoxS1). The Gal4-DBD alone can bind DNA at *GAL4* elements but cannot activate transcription [38,46]. We examined whether the fusion protein could activate expression of two reporter genes, *HIS3* and *MEL1,* in the yeast heterologous system (Figure 2). *HIS3* is essential for histidine metabolism in yeast auxotrophic for histidine, and *MEL1* encodes for alpha galactosidase and results in blue color [46,47]. In this reporter system, *HIS3* and *MEL1* can only be activated if SmSoxS1 functions as a transcriptional activator. We expressed Gal4 DBD-SmSoxS1 fusion protein in yeast and found that the fusion protein was able to activate transcription, resulting in yeast viability on media without histidine (Figure 2A). The Gal4 DBD-SmSoxS1 fusion was also able to activate transcription of MEL1, whose secreted protein product, Mel1 (alpha-galactosidase), produces blue color in the presence of X-alpha-Gal (Figure 2B). Taken together, these data support the role of SmSoxS1 as a functional transcriptional activator.

### 3.4. SmSoxS1 Binds to Sox-Specific Sequences

Sox family proteins are able to bind to DNA through the conserved HMG domain at the Sox consensus binding sequence, (5′-CWTTGWW-3′) [5,11,14,43,48]. Sox2 is able to activate a number of target genes, several of which have potential homologs in schistosomes, including fibroblast growth factor 4 (Fgf4, Smp_035730), Cyclin dependent kinase 5 (Cdk5, Smp_073340), Paired box protein 6 (Pax6, Smp_160670), and Notch Receptor 1 (Notch1, Smp_311360) [5,9,14,45,49,50]. We were able to identify sequences that correlate to the Sox binding consensus in the promoter of these schistosome homologs, as well as in the upstream activation sequence of *SmSOXS1* itself. The observation of the Sox consensus binding sequence in the *SOX2* gene promoter is also seen in other organisms as Sox2 can induce its own gene expression [11,51,52,53,54]. FGF4 is a Sox2 target that is important for embryonic development and cell differentiation and is well studied in terms of Sox2 binding [14,43,49,50,55,56]. To test whether SmSoxS1 can bind to DNA sequences matching the Sox binding elements found in the promoters of *S. mansoni SOXS1* and *FGF4*, we purified the SmSoxS1 protein and performed an electrophoretic mobility shift assay (EMSA). EMSA analysis demonstrated that SmSoxS1 could specifically bind to both Sox binding elements from its own promoter (ACATTGAT) and from the *FGF4* promoter (CTTTGTT) but did not recognize a nonspecific binding sequence, the Ndt80 binding element [57,58] (Figure 3). When the binding affinity of Sox consensus sequences upstream of the *SmSOXS1* gene and upstream of the *FGF4* gene was compared to that of the SmSoxS1 protein, we found that SmSoxS1 has stronger binding affinity to the upstream sequence of *SmSOXS1*, although it could effectively bind to *FGF4* gene Sox consensus (Figure 3).

### 3.5. SmSoxS1 Protein Is Expressed in Cercariae and in 4 h Schistosomula

A custom antibody was raised against the peptide sequence CTNKVVLQHKVKTTS in SmSoxS1 (GenScript). This peptide sequence is specific to SmSoxS1 and is C-terminal to the SmSoxS1 HMG domain. To test the specificity of the antibody, SmSoxS1 was fused with the maltose binding protein (MBP–SmSoxS1) and purified. SmSoxS1 has a predicted size of approximately 33 kDa alone, and, when fused to MBP, the protein is predicted to be about 78 kDa. We detected two bands after protein purification, one at approximately 80 kDa and another around 60 kDa (Figure 4A). When the SmSoxS1 antibody pre-adsorbed with the antigen peptide was used to detect SmSoxS1, both bands disappeared, indicating that both bands are specific to SmSoxS1 and are likely to be a breakdown product of the MBP–SmSoxS1 fusion protein (Figure 4A). The antibody was also tested against protein extract from cercaria, 4 h schistosomula, and 24 h schistosomula (Figure 4B). We found bands at between 33 and 35 kDa in the cercaria and 4 h schistosomula samples and no band at that weight in the 24 h schistosomula (Figure 4B). Another band was found in all samples at just under 40 kDa. When challenged with peptide-pre-adsorbed antibody, both of the 35 kDa bands disappeared, showing that the bands contained SmSoxS1 (Figure 4B). Thus, we find that SmSoxS1 protein was present in cercariae and 4 h schistosomula but was not observed in 24 h schistosomula, correlating with the expression profile observed by quantitative PCR (Figure 1).

### 3.6. SmSoxS1 Localizes to the Schistosomula Anterior Terminus

Using immunohistochemistry, we explored whether SmSoxS1 had any potential localization pattern in cercariae and in 4 h schistosomula. Since SmSoxS1 is not expressed in 24 h schistosomula, it was used as a negative control. We found a significant signal for SmSoxS1 protein localized to the anterior region in cercaria (Figure 5A–D) and in 4 h schistosomula (Figure 5E–H and Figure 6E,F,M,N). In cercariae, SmSoxS1 appeared to be present as punctate staining throughout the head and tail but with distinct and intensified localization at the anterior tip of the cercarial head and at the cercarial neck, or rather the connection point between the cercarial head and tail (Figure 5). With the exception of the posterior staining at the neck, 4 h schistosomula has localization patterns similar to cercariae. SmSoxS1 was not detected in 24 h schistosomula (Figure 5I–L), which correlates with our Western blot analysis and transcript expression profile. When probed with peptide pre-adsorbed antibody, the signal disappears, indicating the signal is specific to SmSoxS1 (Figure 5). Increased magnification to 100× confirmed both the punctate staining and the focused staining at the anterior tip in schistosomula (Figure 6).

### 3.7. Schistosomes Have Multiple Sox Proteins

We initially identified SmSoxS1 as a homolog of Sox2 from mouse based on the BLAST analysis using NCBI and SchistoDB [24,25,26,29]. With the addition of WormBase ParaSite as a database, we identified an additional six potential Sox proteins [25,26,27,28]. All six of these potential schistosome Sox proteins contained a conserved HMG domain [30]. As mentioned earlier, the HMG domain is a key characteristic of the Sox family and is used to classify Sox proteins into their groups: A–J [11,20,21]. Using the HMG domain, we generated a phylogenetic tree to classify known Sox proteins and the schistosome Sox homologs. We included Sox proteins from vertebrates and invertebrates (*Mus musculus*, *Drosophila melanogaster*, *Caenorhabditis elegans*, *Danio rerio*, and *Schmidtea mediterranea)* to ensure that a broad range of life species were used in creating this phylogenetic tree (Appendix A). Consequently, as we expanded the species included and with the additional six schistosome Sox homologs, SmSoxS1 no longer grouped with SoxB proteins but branched off into an earlier group before the SoxB group branch (Figure 7). SmSoxS1 also branches off before the SoxG group separates out from the SoxB group. The additional six schistosome proteins were named SmSoxS2–7. SmSoxS2 and SmSoxS3 grouped with SoxB proteins, while SmSoxS4 grouped with SoxD proteins (Figure 7). The remaining three schistosome sox proteins, SmSoxS5, -S6, and -S7, align with a distinct branch that groups with three planarian Sox proteins (SmdSOXP-1, -2, and -4). This group separated from the other branches early in the phylogenetic tree and reflects a Sox group containing only flatworm proteins (Figure 7).

## 4. Discussion and Conclusions

In an effort to explore the role of Sox in schistosome development, we identified a schistosome homolog (Smp_301790) of mammalian Sox2, which we have named *SmSOXS1* [24,25,26]. The Sox family of proteins frequently function as transcriptional activators and are defined by the sequences of the conserved HMG domain, a DNA binding region that interacts with the Sox DNA element [5,11,14,21,43,59]. *SmSOXS1* is a single exon transcript, around 300 amino acids long, and contains an HMG domain, with 71% identity with the mouse and human Sox2 HMG domain. In addition to the HMG domain homology, the amino acid length and intron-exon structure resemble those of Sox2 in mouse and humans [11,21,59].

Sox proteins bind to DNA at the consensus sequence, (5′-CWTTGWW-3′) [5,11,14,43], and can activate transcription of target genes, such as fibroblast growth factor 4 (FGF4), the transcription factor NeuroD, Cyclin dependent kinase 5 (Cdk5), Paired box filament 6 (Pax6), the filament protein Nestin, Notch receptor 1 (Notch 1), and itself [5,9,14,45,49,50,60,61,62]. When we tested for SmSoxS1 transcriptional activity in a yeast heterologous system as a fusion protein with the Gal4-DBD, SmSoxS1 was able to induce the expression of several yeast reporter genes, demonstrating that, as with many other Sox proteins, SmSoxS1 is a transcriptional activator (Figure 2).

We found Sox binding sequences upstream of several schistosome homologs of Sox2 target genes, including a homolog of fibroblast growth factor 4 (*FGF4*, Smp_035730). We also found the sox binding sequence upstream of *SmSOXS1*. The *SmSOXS1* promoter contains two Sox regulatory regions (SRR), SRR1 and SRR2; the latter, SRR2, contains a Sox binding site. Sox2 and its partner protein recognize SRR in the promoter region of Sox2 and are involved in the expression of Sox2 [51,52,53]. Sox proteins and their partner proteins are able to recognize the Sox and the partner protein binding sites in the promoters of Sox gene targets, and sequence-specific binding determines how target gene expression is regulated [50,63]. SmSoxS1 was able to independently bind to both the sox binding sequences found in the promoter regions of SmSoxS1 and SmFGF4, with preferential binding to the sequence from its own promoter (Figure 3). 

Sox family proteins tend to have partner proteins when binding to DNA, and POU domain proteins are frequently partner proteins for Sox2 [9,14,45,49,51,52,64]. The FGF4 enhancer sequence generally allows for binding of both POU2F1-Sox2 and POU5F1-Sox2 complexes, but only the POU5F1 can complex with Sox2 at SRR2 [49,51]. However, schistosomes do not have an *SmPOU5F1* gene. Regardless, schistosomes have several POU family homologs, including POU2F2 (Smp_122920), POU3F1 (Smp_344230), POU4F1 (Smp_340200), POU4F3 (Smp_000690), and POU6F1 (Smp_139940). While POU5F1 is only found in vertebrates, the other POU family homologs in schistosomes can potentially partner with different Sox proteins [63,65]. The preferential binding of SmSoxS1 to the *SmSOXS1* promoter sequence could be due to the slight variation in the consensus sequence at *SmFGF* gene sequences or potentially a reflection of the previously observed cooperative binding of Sox2 with a partner protein to better bind to the *FGF* promoter. SmSoxS1-specific binding to sox binding sequences further supports its role as a Sox family protein.

After cloning *SmSOXS1*, we explored its expression profile over the schistosome life cycle (Figure 1). Its highest transcript level was in 4 h schistosomula; transcript levels decreased over 3 days and were not detectable in adult schistosomes (Figure 1). The temporal expression pattern of this Sox was of interest as Sox proteins are primarily found in embryos and stem cells and function to maintain pluripotency and various early development processes [5,9,10,11,12,13,14,15,45]. In addition to *SmSOXS1* transcript, other developmental genes are also expressed, such as the schistosome homolog of the Myocyte Enhancer Factor (SmMef2), a member of the conserved MADS-box family of transcription factors. *SmMEF2* is expressed highly in schistosomula, although its expression continues in adults [38,66]. Mef2 proteins have a wide variety of developmental function in vertebrates, including myogenesis, muscle differentiation, heart and craniofacial development, bone development, muscle regeneration, and influencing the immune response. In *Drosophila*, Mef2 is necessary for the development of all muscle cell lineages [66,67]. Similarly, we have unpublished data suggesting that Forkhead box proteins (Fox proteins, in prep) are also expressed during this early stage of schistosome development. Taken together, schistosomes express multiple classes of early developmental genes in the schistosomula. Having ~900 cells, the schistosomula is not defined as an embryonic stage, but schistosomes maintain a class of stem cells that function in the regular turnover of the tegument [68].

Next, we determined SmSoxS1 localization. SmSoxS1 was present in cercaria and 4 h schistosomula, but no protein was found in 24 h schistosomula, reflective of the short steady-state levels of *SmSOXS1 RNA*. SmSoxS1 in cercariae and schistosomula has both low RNA steady state levels and a short temporal time frame (Figure 1). In general, Sox2 has one of the shortest half-lives of the Yamanaka factors, although it is not clear if this is consistent throughout sox genes in general [69].

In cercaria, SmSoxS1 protein is found throughout the head and tail but is concentrated to the anterior end, with some protein found in the region where the tail connects to the head (Figure 5). In 4 h schistosomula, it is mainly found in the anterior region (Figure 5 and Figure 6). We can speculate that this refers to areas of potential stem cell localization. However, while this is a possibility, localization at the anterior and posterior ends of the schistosomulum does not directly overlap with previously identified larvally derived stem cells in schistosomula [70]. Alternatively, these are also regions of the cercaria and 4 h schistosomula where potential damage to the parasite could occur during release of either escape glands, which are thought to assist in exit from the snail, or release of acetabular contents upon host tissue invasion and the separation of the head from the tail [71]. Sox genes play a central role in regenerative wound/injury repair and tissue migration in mice [72]. Thus, it is feasible to speculate that SmSoxS1 protein could function at both these sites of potential new growth and tissue damage.

We initially identified SmSoxS1 based on the initial BLAST analysis and classified SmSoxS1 as a homolog of Sox2. The sequence comparison using NCBI and SchistoDB showed a high degree of homology between the HMG domains of SmSoxS1 and mouse Sox2 [24,25,26,29]. As mentioned, sox family proteins are classified based on the homology of the HMG domain, and proteins of the same group share at least 70% homology [5,11,21]. The HMG domain of SmSoxS1 was 71% identical to that of the mouse Sox2 protein. After the development of WormBase ParaSite, more Sox proteins were identified in schistosomes [25,26,28,73]. In order to classify the new Sox protein, we created an expanded phylogenetic tree using the HMG domain of Sox proteins from *Mus musculus*, *Drosophila melanogaster*, *Caenorhabditis elegans*, *Danio rerio*, and *Schmidtea mediterranea* (Figure 7 and Appendix A). The annotated Sox proteins grouped as expected, whereas classification of schistosome SmSoxS1 was distinct, being more closely related to SoxGs.

As with the SoxG group, SmSoxS1 appears to branch out before the SoxB group forms but earlier than when the SoxG group forms (Figure 7) [21,45,74]. While mammalian Sox15, the single gene in the SoxG group, clusters near the SoxB group, its function is distinct from Sox2 and other SoxBs [61,74,75]. Sox15 is expressed in mouse embryonic cells along with Sox2, but, unlike Sox2, Sox15-null cells are viable [61]. Sox15 is also not involved in pluripotency but appears to be a part of skeletal muscle regeneration and helps regulate myogenetic progenitor cells [75,76]. Finally, Sox15 is the only member of the SoxG group and is only found in mammals [61], which makes the proximity of SmSoxS1 and Sox15 more intriguing as it has the potential to be more related to the SoxG group of Sox proteins [61]. However, because SmSoxS1 does not directly cluster with SoxG on the phylogenetic tree, we chose not to place it overtly in the SoxG group.

As part of this phylogenetic analysis, we included gene sequences from a broad variety of species, including the free-living flatworm planarian, *Schmidtea mediterranea*, and the roundworm *C elegans*. SmSoxS and SmSoxS3 are grouped as SoxB proteins and SmSoxS4 was classified as a Sox D protein (Figure 7). Based on the sequence in the database, SmSoxS4 has an intron in the HMG domain. SoxD proteins are known for having an HMG domain intron, and the intron in SmSoxS4 is located near the conserved location for SoxD HMG domain introns [21]. The presence of this intron supports SmSoxS4 being classified as a SoxD. No schistosome sox homologs were classified as Sox C, E, or F groups.

One of the more interesting parts of the phylogenetic tree is the grouping of an only flatworm proteins clade. This group consists of three schistosome and three planarian proteins and branches out earlier than other branches in the phylogenetic tree (Figure 7). The transcript expression of one of these, which we name SmSox7, was initially characterized as a sox14. Our data suggest that it belongs to a unique class of flatworm-specific sox proteins instead. While the Sox family has been well studied in vertebrates, there has been less exploration on sox proteins in invertebrates, especially flatworms [20,45,77,78,79]. Further exploration of the role of Sox in parasitic schistosomes could enhance our knowledge of schistosome development and function as a potential drug target against helminth diseases. Currently, the work in *S. mediterranea* best represents the efforts in any flatworm for understanding sox genes and serves as a complementary model to the work in schistosomes for understanding some of the basic biology of flatworm development and whether sox genes have remained consistent or evolved alternative roles in parasitic versus free-living flatworms.

## Figures and Tables

**Figure 1 pathogens-12-00690-f001:**
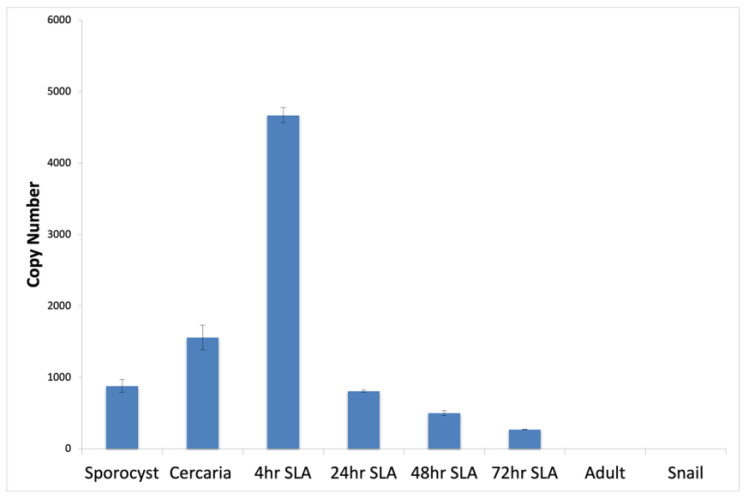
Absolute quantitative PCR for the schistosome SOXS1 transcript during schistosome development.

**Figure 2 pathogens-12-00690-f002:**
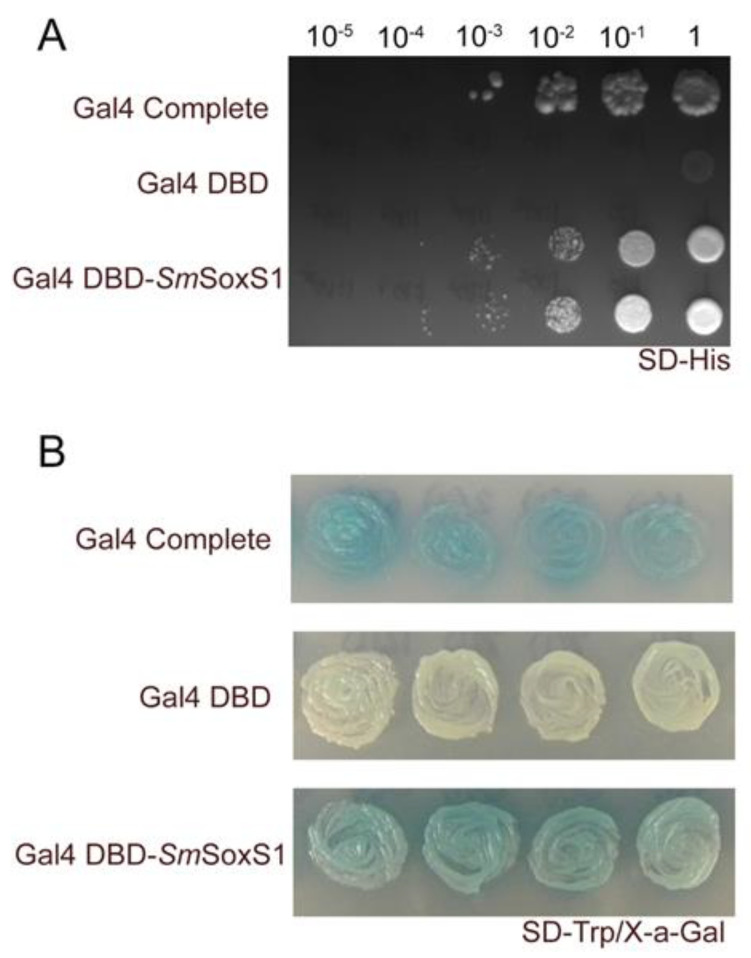
SmSoxS1 is a transcriptional activator. SmSoxS1 was fused to the DNA binding domain of the yeast Gal4 protein (Gal4-DBD) and expressed in yeast cells. (**A**) A serial dilution (ranging from 1 to 10^−5^) was plated on SD-Histidine plates to select for yeast cells that induce HIS3 expression. (**B**) An alpha galactosidase assay with the yeast MEL1 reporter gene was used as a secondary screen for transcriptional activation. Blue color change indicates gene activation, while the negative control remains white. The yeast activator Gal4 was used as a positive control, and the Gal4 DNA binding domain (Gal4-DBD) was used as a negative control.

**Figure 3 pathogens-12-00690-f003:**
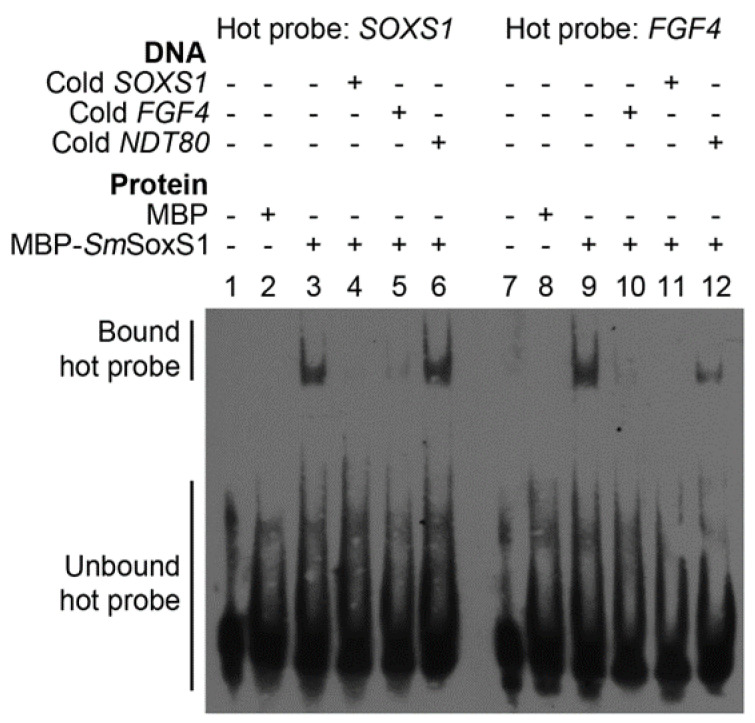
SmSoxS1 targets DNA sequences containing the Sox binding element. Biotin-labeled DNA probes containing the Sox binding element upstream of SmSOXS1 (lanes 1–6) and SmFGF4 (lanes 7–12) were incubated with the SmSoxS1 recombinant protein with or without competitor DNA. Binding reactions contained 200 fmol labeled DNA probe, 200 pmol competitor DNA (1000×), 17.6 µg MBP, or 12.4 µg recombinant SmSoxS1 protein.

**Figure 4 pathogens-12-00690-f004:**
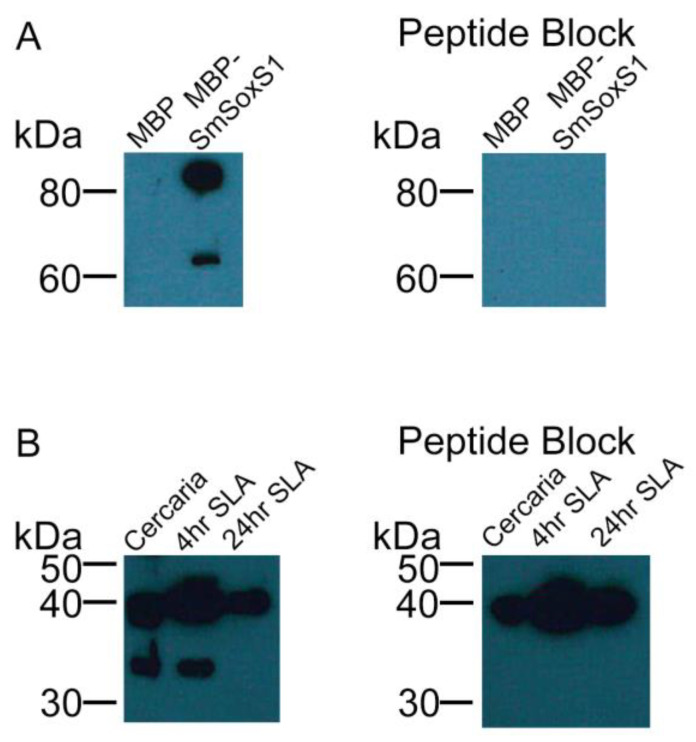
A custom SmSoxS1 antibody recognizes SmSoxS1. (**A**) Western blot analysis using an SmSoxS1 antibody detected MBP–SmSoxS1 recombinant protein (50 ng) expressed in bacteria. 50 ng of MBP is used as a negative control. (**B**) Western blot analysis using an SmSoxS1 antibody detected SmSoxS1 with 10ug of protein extract from cercaria, 4 h schistosomula, and 24 h schistosomula. The peptide block, a pre-adsorption using the SmSoxS1 antigen at a 10× concentration by weight, was used to test for antibody specificity in both Western blots.

**Figure 5 pathogens-12-00690-f005:**
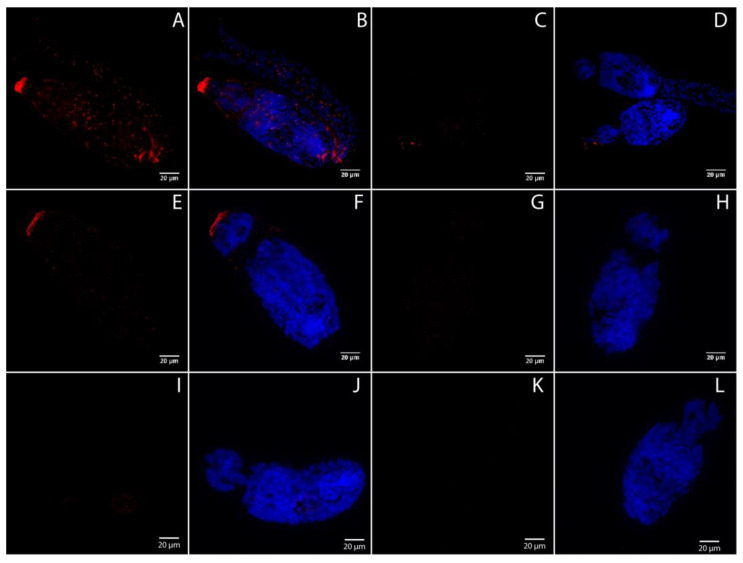
SmSoxS1 is present in cercariae and 4 h schistosomula but absent in 24 h schistosomula. Representative confocal maximum projections of cercariae, 4 h, and 24 h schistosomula. Anti-SmSoxS1, as well as a peptide block pre-absorption using the SmSoxS1 antigen, and Alexa 647 secondary were used to detect SmSoxS1 localization in different life stages. All the images were taken with 63× objective. (**A**–**D**) show cercariae. (**A**) SmSoxS1 (red) signal alone; (**B**) SmSoxS1 and DAPI (blue) overlay. (**C**) SmSoxS1 signal alone with pre-absorption; (**D**) DAPI and SmSoxS1 overlay with pre-absorption. (**E**–**H**) show 4 h schistosomula. (**E**) SmSoxS1 signal alone; (**F**) SmSoxS1 and DAPI overlay. (**G**) SmSoxS1 signal alone with pre-absorption; (**H**) DAPI and SmSoxS1 overlay with pre-absorption. (**I**–**L**) show 24 h schistosomula. (**I**) SmSoxS1 signal alone; (**J**) SmSoxS1 and DAPI overlay. (**K**) SmSoxS1 signal alone with pre-absorption; (**L**) DAPI and SmSoxS1 overlay with pre-absorption.

**Figure 6 pathogens-12-00690-f006:**
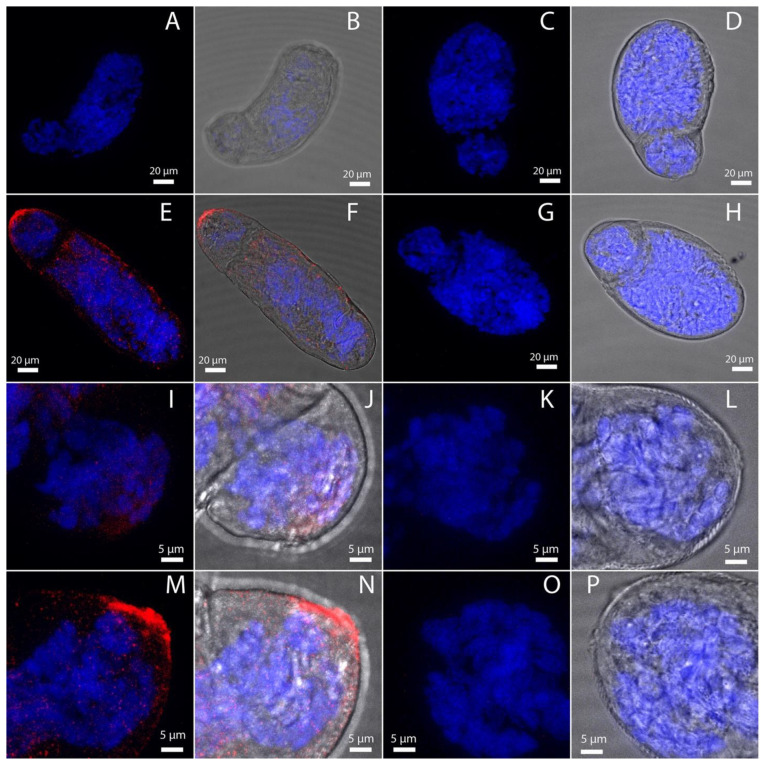
SmSoxS1 is localized to the anterior tips of early developing schistosomula. Representative confocal maximum projections of 4 h and 24 h schistosomula at 67× (**A**–**H**) and 100× (**I**–**P**). Anti-SmSoxS1 antibody and a peptide block pre-absorption using the SmSoxS1 antigen with Alexa 647 secondary was used to detect SmSoxS1 localization. (**A**–**D**) show the pre-absorption for 4 h and 24 h schistosomula at a 63× objective. (**A**) SmSoxS1 (red) and DAPI (blue) signal for 4 h schistosomula, (**B**) SmSoxS1, DAPI, and BF overlay for 4 h schistosomula, (**C**) SmSoxS1 and DAPI overlay for 24 h schistosomula, and (**D**) SmSoxS1, DAPI, and BF overlay for 24 h schistosomula. (**E**–**H**) show the experimental localization of SmSoxS1 at a 63× objective. (**E**) SmSoxS1 and DAPI signal for 4 h schistosomula, (**F**) SmSoxS1, DAPI, and BF overlay for 4 h schistosomula, (**G**) SmSoxS1 and DAPI overlay for 24 h schistosomula, and (**H**) SmSoxS1, DAPI, and BF overlay for 24 h schistosomula. (**I**–**L**) show the pre-absorption for 4 h and 24 h schistosomula at a 100× objective. (**I**) SmSoxS1 and DAPI signal for 4 h schistosomula, (**J**) SmSoxS1, DAPI, and BF overlay for 4 h schistosomula, (**K**) SmSoxS1 and DAPI overlay for 24 h schistosomula, and (**L**) SmSoxS1, DAPI, and BF overlay for 24 h schistosomula. (**M**–**P**) show the experimental localization of SmSoxS1 using 100× objective. (**M**) SmSoxS1 and DAPI signal for 4 h schistosomula, (**N**) SmSoxS1, DAPI, and BF overlay for 4 h schistosomula, (**O**) SmSoxS1 and DAPI overlay for 24 h schistosomula, and (**P**) SmSoxS1, DAPI, and BF overlay for 24 h schistosomula.

**Figure 7 pathogens-12-00690-f007:**
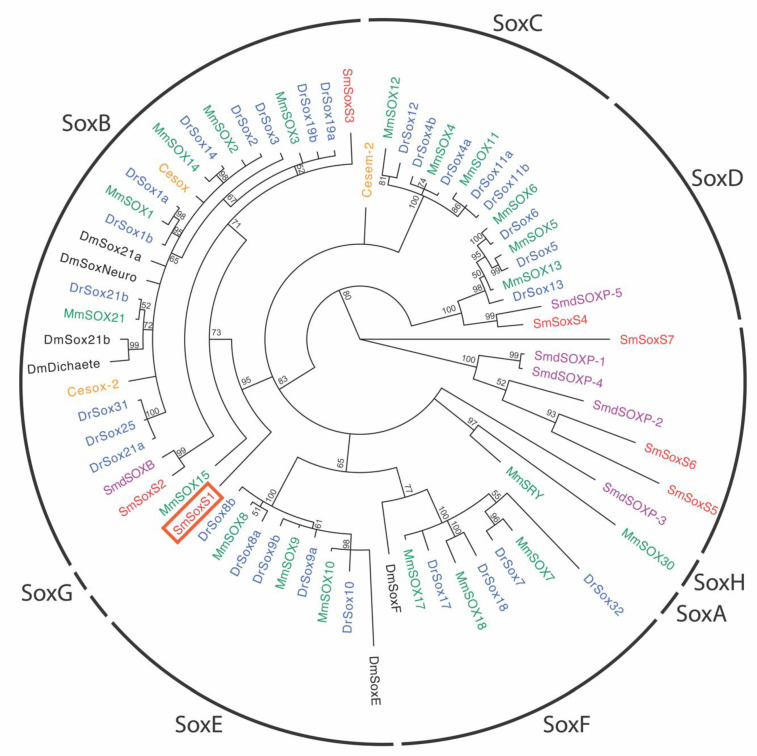
Phylogenetic tree of the protein sequence of the HMG domain of Sox proteins. The phylogenetic tree was generated using MrBayes with 3,000,000 generations and displayed using FigTree. Posterior probability is 100 unless marked next to node. SmSoxS1 is boxed. Abbreviations are as follows: *Schistosoma mansoni* (Sm), *Mus musculus* (Mm), *Drosophila melanogaster* (Dm), *Caenorhabditis elegans* (Ce), *Danio rerio* (Dr), and *Schmidtea mediterranea* (Smd). Accession numbers are located in Appendix A.

## Data Availability

No new data were created or analyzed in this study. Data sharing is not applicable to this article.

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
