# Peer review of "Characterization of Schistosome Sox Genes and Identification of a Flatworm Class of Sox Regulators"

_pathogens, 2023, doi:10.3390/pathogens12050690_

Round 1

Reviewer 1 Report

Presented manuscript describes the comprehensive characterization of Sox genes of S. mansoni using of various methods/approaches including phylogenetic analysis, qPCR, Immunohistochemistry, confocal imaging, functional testing of recombinant protein. I found the text clearly written with lot of valuable results and outcomes. I have couple minor comments which I divided according  to article breakdown.

Introduction:

The introduction is well written with the enough information for further understanding of the manuscript. However, I have couple of minor comments. 

Introduction is heavily focused on function of mammalian Sox genes (indeed – there is lot of information due to the intensive research). Authors mention characterization of Sox2 function in Caenorhabditis elegans and Drosophila melanogaster. I would consider mentioning the specific functions, which would be more informative than information about e. g. taste bud development in mammals. 

Line 36-37 „In many organisms embryonic development requires the expression of a host of embryonic genes” - unclear to me, do you mean parasitic organisms? They need the expression of host embryonic genes for they own development? 

Materials and methods:

Materials and methods are clearly written enabling to repeat the experiments, just minor comments below

Line 77 – 78 I would consider briefly adding the method of transformation, e. g. S. mansoni cercariae were collected and MECHANICALY transformed into schistosomula as previously described. 

Line 83-84 schistoDB.net seems to me retired and not accessible. This is the message when trying to connect. “SchistoDB.net has been retired. We encourage you to use WormBase ParaSites to access your genomes of interest“. It would be also nice to provide the version of the genome e.g Assembly SM_V10, database version WBPS18

Line 130 - E. coli should be in italic 

Results:

Results are well organised and understandable, just minor comments.

Line 216 – wasn’t it 4h schistosomula instead of 3h schistosomula?

I would be consistent in terminology, in the manuscript there are at least 3 version of the age of schistosomula - 3h schistosomula, 4-hour schistosomula, 4hr schistosomula (figure 1.)

Figure 1. I would not repeat “schistosomula” four times, e.g it could be one word for all timepoints or just abbreviated. 

            - There are error bars in the chart, however I didn’t find the information about the number of replicates, must be in methods.

Discussion:

Comprehensive, well linked with the known literature.

 Only one typo 

Line 456 - or release oF acetabular 

Reviewer 2 Report

Generally, the authors reported their work on identification of new sox-like genes in S. mediterranea, which also has the potentiality to become a new criteria in Planarian classification. Undoubtedly, this manuscript can meet the requirement of special issue: "The Parasitic Schistosome Worm and Schistosomiasis" and will be of interest to the readers of Pathogens. While I think this work can be accepted, I still have some suggestions on this manuscript to make it easier to understand by potential readers.

1. Page 5, line 207: The authors claimed sequence similarity of mSOX1, hSOX1 and SmSoxS1. It will be more clearly to show if the authors can add alignment results of the 3 proteins. Also I did not quite understand why the authors only talked about sequence homology only with first 100 AA of mSOX1 and SmSoxS1. The authors did try to address that conserved HMG domain is in the first 100 AA, but partial similarity is not enough to convince people that these proteins are homology proteins. Hence, the authors should make a clarification here for the reason.

2. Page 7, line 283: The authors did not include the sequence of the raised Ab. The sequence of SmSoxS1 is not included either. It's more convenient to reproduce the work if the authors can include the sequence information in supplementary files.

3. For figure 2-4, a larger and higher resolution of figures should be applied. It's difficult to determine the detail of these figures, especially figure 2 and 4.

Some typos appeared in this manuscript. For example: Abstract, line 19: Planerians should be Planarians.

The overall English language quality is good, but the authors are suggested to double-check the typos in the manuscript.
